# The ultrafast onset of exciton formation in 2D semiconductors

Chiara Trovatello[1], Florian Katsch [2], Nicholas J. Borys [3,4], Malte Selig[2], Kaiyuan Yao[3,5,6], Rocio Borrego-Varillas[1,7], Francesco Scotognella[1,3], Ilka Kriegel[3,8], Aiming Yan [9,10,11,12], Alex Zettl [9,10,11], P. James Schuck [3,5,6], Andreas Knorr[2 ✉], Giulio Cerullo [1,7 ✉] & Stefano Dal Conte [1 ✉]

The equilibrium and non-equilibrium optical properties of single-layer transition metal dichalcogenides (TMDs) are determined by strongly bound excitons. Exciton relaxation dynamics in TMDs have been extensively studied by time-domain optical spectroscopies. However, the formation dynamics of excitons following non-resonant photoexcitation of free electron-hole pairs have been challenging to directly probe because of their inherently fast timescales. Here, we use extremely short optical pulses to non-resonantly excite an electron-hole plasma and show the formation of two-dimensional excitons in single-layer $MoS_2$ on the timescale of 30 fs via the induced changes to photo-absorption. These formation dynamics are significantly faster than in conventional 2D quantum wells and are attributed to the intense Coulombic interactions present in 2D TMDs. A theoretical model of a coherent polarization that dephases and relaxes to an incoherent exciton population reproduces the experimental dynamics on the sub-100-fs timescale and sheds light into the underlying mechanism of how the lowest-energy excitons, which are the most important for optoelectronic applications, form from higher-energy excitations. Importantly, a phonon-mediated exciton cascade from higher energy states to the ground excitonic state is found to be the rate-limiting process. These results set an ultimate timescale of the exciton formation in TMDs and elucidate the exceptionally fast physical mechanism behind this process.

[1] Dipartimento di Fisica, Politecnico di Milano, Piazza L. da Vinci 32, Milano I-20133, Italy. [2] Institut für Theoretische Physik, Technische Universität Berlin, 10623 Berlin, Germany. [3] Molecular Foundry, Lawrence Berkeley National Laboratory, Berkeley, CA 94720, USA. [4] Department of Physics, Montana State University, Bozeman, MT, USA. [5] Department of Mechanical Engineering, University of California, Berkeley, CA 94720, USA. [6] Department of Mechanical Engineering, Columbia University, New York, NY 10027, USA. [7] IFN-CNR, Piazza L. da Vinci 32, Milano I-20133, Italy. [8] Functional Nanosystems, Istituto Italiano di Tecnologia (IIT), via Morego, 30, Genova 16163, Italy. [9] Department of Physics, University of California at Berkeley, Berkeley, CA 94720, USA. [10] Materials Sciences Division, Lawrence Berkeley National Laboratory, Berkeley, CA 94720, USA. [11] Kavli Energy NanoSciences Institute at the University of California, Berkeley and the Lawrence Berkeley National Laboratory, Berkeley, CA 94720, USA. [12] Present address: Department of Physics, University of California at Santa Cruz, Santa Cruz, CA 95064, USA. ✉email: andreas.knorr@tu-berlin.de; giulio.cerullo@polimi.it; stefano.dalconte@polimi.it

Single-layer (1L) TMDs are attracting growing interest because of their peculiar properties that make them highly suitable for optoelectronics applications[1]. The reduced dielectric screening that is caused by the strong spatial confinement results in an optical response that is dominated even at room temperature by strongly bound excitons with large (hundreds of meV) binding energies[2,3]. The enhanced Coulomb interaction gives rise to additional effects such as the occurrence of a Rydberg series of excitonic states[4] and many-body complexes like trions[5] and biexcitons[6] whose physical properties can be tuned by changing the dielectric environment[7] or by applying external stimuli such as light[8], strain[9], electric[10], and magnetic fields[11]. Ultrashort laser pulses offer an additional way to change and potentially control the properties of excitons on a fast timescale. The non-equilibrium optical response of TMDs has been extensively explored both experimentally and theoretically[12,13]. The dynamical response of TMDs and, in general, all semiconductors can be divided into coherent and incoherent regimes[14,15]. While the incoherent exciton dynamics of TMDs, including processes, such as exciton thermalization[16], radiative/non radiative recombination[17], intra and intervalley scattering[18–22], exciton dissociation[23], and exciton–exciton annihilation[24], have been extensively studied, the coherent exciton response and the corresponding early stage dynamics of the exciton formation process are still almost unexplored largely because the limited available temporal resolution (i.e., >100 fs for pump-probe optical spectroscopy and >1 ps for time-resolved photoluminescence) has prevented the study of the primary early stage dynamics of exciton photo-generation processes[17,25,26].

Here, we push the temporal resolution of ultrafast differential reflectivity (ΔR/R) spectroscopy to the regime of less than 30 fs in order to investigate the dynamics of exciton formation in 1L-MoS$_2$[27]. Experimentally, we employ a pump-probe technique that uses ΔR/R to monitor the evolution of the excited state population. Particularly, we elucidate how long it takes for an initial population of high-energy photoexcited electrons and holes to relax to the lowest-energy exciton states (i.e., the 1s states of the A and B excitons), and how this formation time depends upon the energy of the initial state (c.f. Fig. 1). The measured dynamics are found to be remarkably fast, suggesting that lowest-energy excitons form from these high-energy populations with a characteristic timescale as fast as 10 fs. Upon increasing the energy of the initial state of photo-injected carriers (i.e., by increasing the pump photon energy), we find that (1) the formation time of the excitons increases monotonically with increasing energy; (2) an initial, sub-100 fs fast decay component of the excitons vanishes; and (3) the dynamics of a slower decay process (on the timescale of picoseconds) associated with the relaxation of a thermal population of excitons does not substantially change. Simulations based on the TMD Bloch equations attribute the excitation energy dependence of the formation dynamics to a phonon-induced cascade-like relaxation process of high-energy incoherent excitons down to the excitonic ground state[28]. This process, although extremely fast, is predicted to have a characteristic time constant between 20 and 30 fs, corroborating our experimental observations. The early sub-100 fs relaxation dynamics are also well captured by the simulations, which reveal that they arise from the decay of the pump-induced coherent optical polarization into ultimately thermal exciton populations (i.e., incoherent excitons). These results are of great relevance for optoelectronic applications of TMDs as they directly define a timescale for an efficient extraction of hot carriers in 1L-TMDs and TMD-based heterostructures before the exciton formation process.

## Results

**Ultrafast exciton dynamics.** Figure 1 schematically summarizes the exciton formation process (Fig. 1a) and the key optical

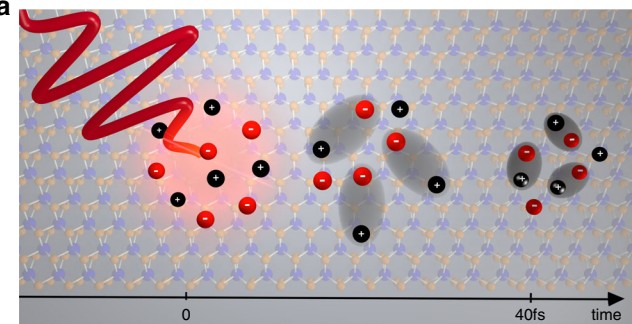

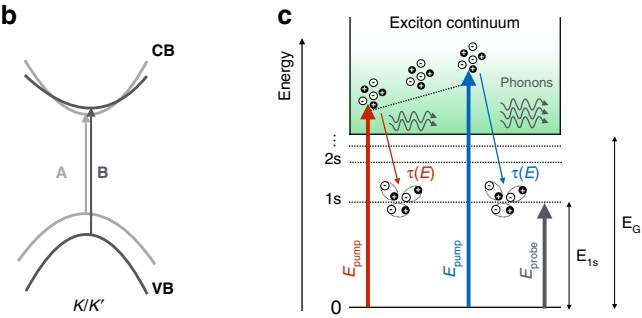

**Fig. 1 Exciton formation process in 1L-MoS$_2$. a** Cartoon of the exciton formation process after photo-injection of free electron-hole pairs. **b** Schematic illustration of the single particle band structure of 1L-MoS$_2$ at the K/K' points. The two arrows represent A/B excitonic transitions, split due to the strong spin-orbit interaction at the K/K' points of the Brillouin zone. **c** Sketch of the pump-probe experiment. A few-optical-cycle laser pulse injects free electron/hole pairs at increasing energies above the exciton continuum (E$_G$). These quasiparticles lose their initial kinetic energy and scatter down via a cascade process mediated by phonons to lower-lying discrete excitonic states until they reach the 1s excitonic state. The timescale τ(E) of this relaxation process is determined by measuring the absorption change of a probe beam, tuned on resonance with the 1s state, due to the Pauli blocking effect.

excitonic transitions (Fig. 1b) of 1L-MoS$_2$. The 1L-MoS$_2$ used here is grown by chemical vapor deposition on a SiO$_2$/Si substrate. All measurements were made on as-grown samples. From the steady-state reflectance measurements that are reported in Supplementary Fig. 1, the lowest energy A and B excitonic resonances that arise from optical transitions between the two-highest energy valence bands and the lowest-energy conduction bands at the K and K' points in the Brillouin zone are centered at the energies of 1.88 eV and 2.03 eV, respectively (see sketch in Fig. 1b). As illustrated in Fig. 1c, both the A and B optical transitions form two manifolds of Rydberg-like series of bound states that merge into a continuum of unbound electron-hole states[3,29]. These resonances correspond to the lowest-energy (or ground state) excitons in the A and B manifolds (i.e., the A$_{1s}$ and B$_{1s}$ states, respectively), possess the largest oscillator strengths, and dominate the optical response over the corresponding higher-energy unbound states. Based on previous experimental studies[3,29] and refined theoretical models[18], we estimate that the exciton binding energy (i.e., the energetic difference between the 1s exciton and the lowest energy state in the exciton continuum) is of the order of 350 meV. In our measurements the pump photon energies span from 2.29 eV to 2.75 eV (with bandwidths spanning from 70 meV to 140 meV) and thus, at all energies, photoexcitation predominantly creates an initial population of excitons at or well-into the exciton continuum for the A excitons (as well as for the B excitons at energies above ~2.4 eV).

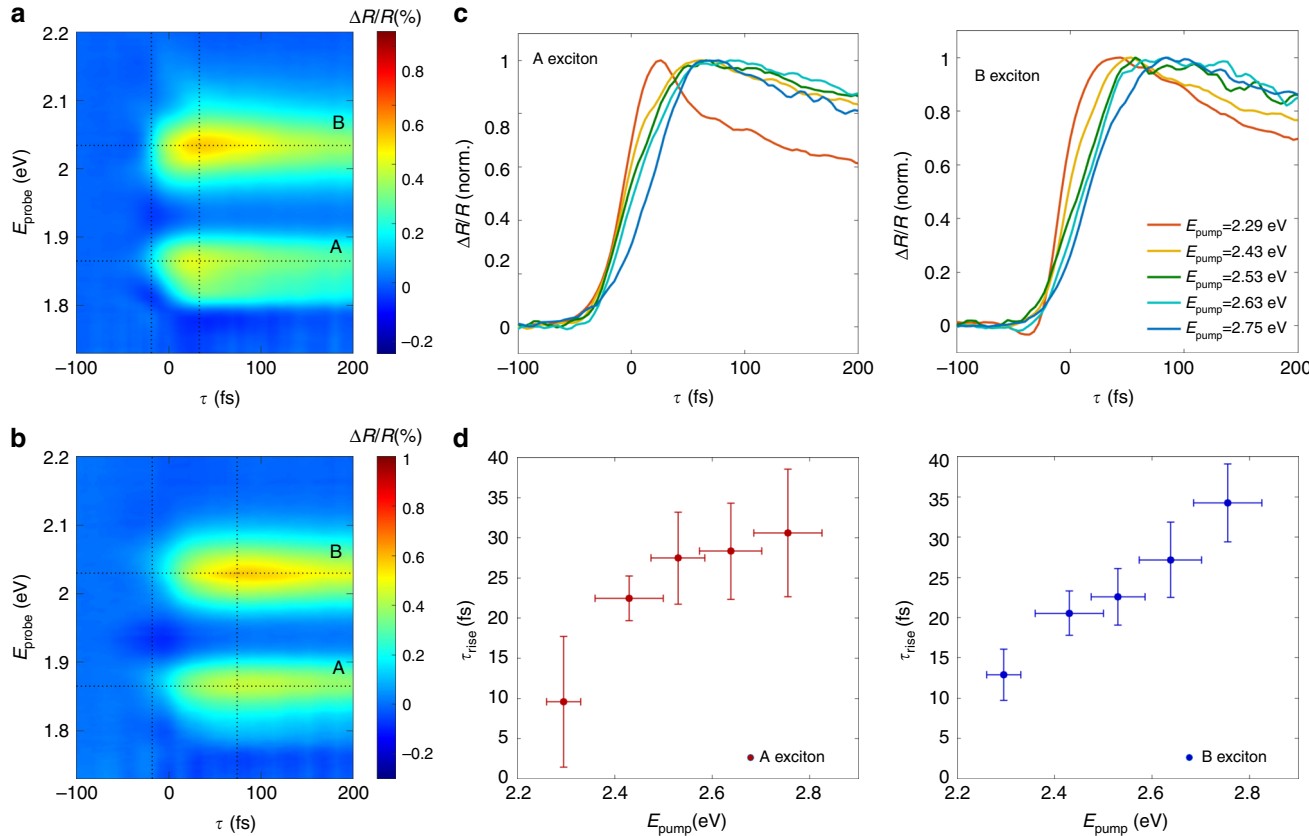

**Fig. 2 Energy-dependent exciton formation process.** ΔR/R maps measured on 1L-MoS$_2$ photoexcited **a** at 2.29 eV and **b** at 2.75 eV. The measurements are performed at room temperature. The horizontal dashed lines pass through the maximum of the ΔR/R spectrum at the energies of the A$_{1s}$/B$_{1s}$ exciton transitions, while the vertical lines mark the temporal range from 10 to 90% of the build-up signal. The excitation fluence is 5 $\mu$Jcm$^{-2}$. Pump and probe beams have parallel and linear polarizations. The time zero is defined, for each measurement, as the maximum of the cross-correlation signal between the pump and the probe pulses as explained in the Methods. **c** Temporal cuts of the ΔR/R maps measured across the A$_{1s}$ and B$_{1s}$ excitonic resonances for increasing pump photon energy. **d** Pump photon energy dependence of $\tau_{rise}$. Horizontal error bars are determined by the bandwidth of the pump pulses; vertical error bars are obtained from the fits of the time traces.

To monitor the formation dynamics and quantify the formation time (i.e., $\tau(E)$ in Fig. 1c), our probe energies monitor the rise and decay dynamics of the ΔR/R signals for the 1s states of the A and B excitons.

We use tunable pump pulses to photo-inject electron-hole pairs with increasing excess energy with respect to E$_G$. The time-delayed broadband probe pulses measure the build-up and early decay dynamics of both A$_{1s}$ and B$_{1s}$ excitons. The experimental temporal resolution of the setup is sub-30-fs (see Methods and Supplementary Fig. 5). Figure 2a and b reports ΔR/R(E$_{probe}$,$\tau$) maps as a function of the pump-probe delay $\tau$ and the probe photon energy E$_{probe}$ for two different pump photon energies: 2.29 eV and 2.75 eV, i.e., respectively, across and well above E$_G$. Due to the presence of the reflecting Si substrate, ΔR/R is equivalent to a double-pass differential transmission (ΔT/T). Both measurements show comparable qualitative responses: positive strong features associated with the A$_{1s}$ (centered at E$_{probe}$ = 1.88 eV) and B$_{1s}$ (centered at E$_{probe}$ = 2.03 eV) exciton states. The ΔR/R spectrum displays a symmetric profile around each excitonic resonance with no shift of the peak maximum, within the considered temporal window (i.e., from −100 to 200 fs). Weak and broad negative features are detected at higher and lower probe energies as better shown in Supplementary Fig. 4. These features are strongly quenched with respect to previously published pump-probe measurements on 1L-MoS$_2$ performed in transmission geometry[12]. We attribute the different intensities of these signals away from the excitonic

transitions to a photonic effect caused by the interference of multiple reflections of the incoming beam in the thin SiO$_2$ substrate.

In the temporal window explored in the experiment (i.e., ~200 fs), the ΔR/R spectrum displays a symmetric profile around each excitonic resonance with no shift of the peak maximum, showing that the signal is dominated by Pauli blocking (see Supplementary Fig. 2).

A closer inspection of the maps reveals that the formation time $\tau_{rise}$ of the ΔR/R signals of the A$_{1s}$ and B$_{1s}$ excitons is longer for higher pump photon energies. When the excitation energy is close to E$_G$, the signal displays a quasi-instantaneous (i.e., pulse-width limited) build-up, while for increased pump photon energy, $\tau_{rise}$ significantly increases. We stress that this result can be observed only thanks to the high temporal resolution of the setup. To better quantify this effect, Fig. 2c reports the temporal cuts of the maps taken at the A$_{1s}$ and B$_{1s}$ exciton peak for increasing pump photon energies. The build-up times are estimated by fitting the temporal traces in the temporal window between −100 and 200 fs, to the function:

$$f \propto (1 - e^{-t/\tau_{rise}}) * H * (A_1 e^{-t/\tau_1} + A_2 e^{-t/\tau_2}) \qquad (1)$$

where H is the Heaviside function centered at $\tau = 0$; $\tau_1$ and $\tau_2$ are decay constants and $A_1$ and $A_2$ are the amplitudes of each decay component. The fitting function is convoluted with a Gaussian profile corresponding to the instrumental response function, which is the experimentally measured cross-correlation profile between the pump and the probe (see Methods and Supplementary Note 5 for

the temporal characterization of the pulses). We stress that only after including a finite exponential rise time in the fitting function, we can satisfactorily reproduce the experimental formation time (see Supplementary Fig. 6). Figure 2d unambiguously shows that $\tau_{rise}$ monotonically increases with the initial excess energy of the photoinjected carriers. Interestingly, for a higher pump energy excitation (i.e., 3.75 eV), we observe that $\tau_{rise}$ sensitively deviates from the increasing energy trend reported in Fig. 2d (see Supplementary Note 3). We can tentatively explain this flattening of the formation dynamics as a result of different phonon-mediated scattering process involving electronic states far away from the K/K' points. The weak negative dip observed for negative times in the B exciton signal for 2.29 eV excitation photon energy is attributed to the so-called pump-perturbed free-induction decay (PPFID)[30]. In this process that occurs when the probe pulse precedes the pump, the free-induction decay field emitted by the sample excited by the probe pulse is perturbed by the interaction with the pump pulse, giving rise to oscillating signals at negative delays. The PPFID effect can be safely disregarded from the measured build-up timescale of the sample because its amplitude is more than one order of magnitude lower than the bleaching of the excitonic peak. Another interesting effect is the extremely fast decay observed at lower pump photon energy, which occurs on a timescale comparable to that of the build-up and fades away as the pump is tuned to higher photon energies. This effect is particularly clear on the A exciton $\Delta R/R$ trace (see Fig. 2c).

**Theoretical model**. To identify the essential mechanisms that underlie the early-time non-equilibrium optical response of 1L-MoS$_2$, we performed simulations based on the TMD Bloch equations[31]. Our model describes the temporal dynamics of the excitons after the photoexcitation of free electron-hole pairs above or close to E$_G$. The $\Delta R/R$ at the A/B exciton transitions due to the photoexcitation process is also calculated and compared with experimental results. Excitonic excitations are theoretically described by solving the Wannier equation (see Supplementary Note 8)[31,32]. The exciton kinetics are determined by a set of coupled differential equations (TMD Bloch equations) describing the temporal evolution of the polarization (in our theory an excitonic scattering state) and the excitonic population (incoherent excitons) where the phonon-mediated relaxation from energetically higher densities is described with effective rates $\Gamma_{\nu+1\rightarrow\nu}$ determined by independent density functional theory calculations (see Methods section and the sketch in Fig. 3a). Solving the set of equations of motion for the coherent polarization as well as incoherent exciton population gives access to the temporal dynamics of the differential reflectance at the 1s exciton resonance frequencies.

The results of the simulations, reported in Fig. 3b, agree remarkably well with the experimental results. For all pump photon energies, the calculated $\Delta R/R$ exhibits sub-100 fs build-up followed by a decay time on the picosecond timescale. At higher pump energies, the peak of the signal is progressively delayed as a result of a slower rise time. The timescale and the pump photon energy dependence of the build-up dynamics are in remarkable quantitative agreement with the experimental results. The slowing of $\tau_{rise}$ with increasing pump photon energy is the result of a high-energy exciton cascade scattering process down to the excitonic ground state. A sketch of the scattering processes involving photoexcited excitons is reported in Fig. 3a. The nonresonant pump pulse induces an almost instantaneous coherent interband exciton polarization which oscillates with the same frequency of the driving pulse. This polarization rapidly dephases by exciton–phonon scattering and leads to a delayed formation of an incoherent exciton population involving electronic states above E$_G$. These high-energy weakly bound excitons

quickly lose their energy and scatter into lower-lying continuum and discrete excitonic states. The scattering process continues until most of the excitonic population reaches the lowest energy 1s exciton state. With increasing pump photon energy, the number of intermediate scattering events, required to complete the cascade process, increases. This increase in needed scattering events results in the experimentally observed delayed formation of the bleaching signal measured at the A$_{1s}$/B$_{1s}$ optical transitions.

The simulations also capture the occurrence of a sub-100 fs decay component, which is particularly evident in the $\Delta R/R$ temporal trace for the A$_{1s}$ state and progressively vanishes at higher pump photon energies. We attribute this effect to an interplay between a coherent exciton polarization and an incoherent exciton density. The coherent contribution adiabatically follows the pump pulse and gives rise to an instantaneous coherent build-up, while the incoherent signal is characterized by a delayed formation time involving additional exciton–phonon scattering processes. Figure 4 compares the full transient signal incorporating both contributions (solid line) as well as the underlying coherent (shaded area) and incoherent (dashed line) contributions to the differential signal obtained by artificially turning off the coherent contribution for low and high pump photon energies. For low pump photon energies, the full signal and the incoherent part of the signal display different dynamics in the first tens of femtoseconds and almost overlap after ~100 fs. While the build-up dynamics are dominated by the coherent excitons, the energetically lowest incoherent exciton densities play a dominant role after the polarization to population transfer is complete. We stress that, in this excitation regime, coherent exciton contribution could be more clearly disentangled from the incoherent exciton dynamics by performing pump-probe measurements that utilized extremely short, linearly, and circularly polarized pulses and polarization resolved detection schemes. However, the use of broadband polarization optics makes it difficult to preserve the extremely high temporal resolution needed to observe such a process.

For high pump photon energies, the difference between full and incoherent curves diminishes since the coherent part inversely depends on the detuning between pump and probe pulses. This difference is related to the polarization, which rapidly decays before a significant 1s exciton density builds up that can be detected by the probe pulse. Thus, the transient signal in this energy regime is strongly dominated by the dynamics of incoherent exciton densities. In this excitation regime, the extracted rise time (~30 fs) of the $\Delta R/R$ traces is a direct estimation of the timescale of the incoherent exciton formation process.

## Discussion

Our combined experimental and theoretical work defines a different timescale for the exciton formation process in 1L-TMDs, which is much faster than the one previously estimated by intraexcitonic mid-IR[33,34] and interband visible optical spectroscopy[25]. We also stress that this exciton formation process is faster than the ~1 ps trion formation time in TMDs[5] and, remarkably, orders of magnitude faster than the formation time of excitons in quantum wells (i.e., ~1 ns) measured by time-resolved photoluminescence[35] and transient terahertz spectroscopy[36]. This result points to a correlation between the formation time and the binding energy of the excitons. The reduced Coulomb screening in TMDs makes this process more rapid and effective than for weakly bound excitons in quantum wells or trions. Two different mechanisms have been proposed to describe the exciton formation process in semiconductors: geminate and bimolecular[37]. In the geminate mechanism excitons are directly

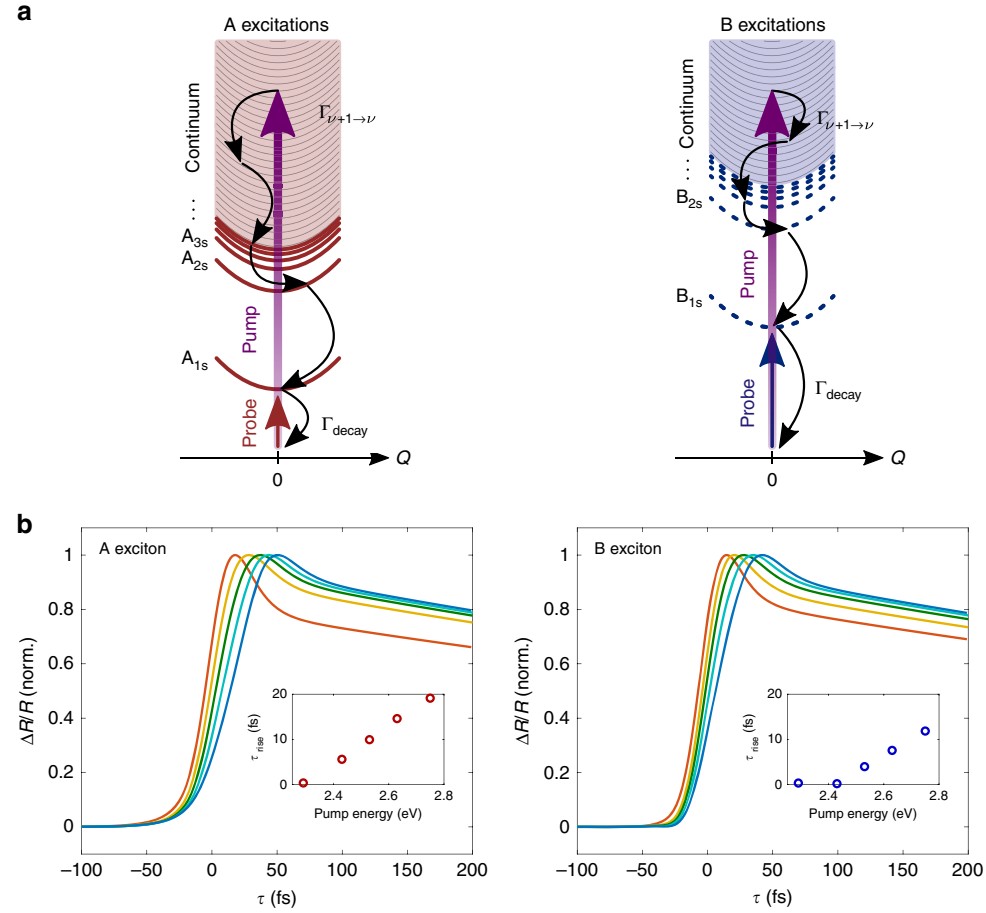

**Fig. 3 Simulation of the exciton formation process. a** Schematic illustration of the relaxation model in the exciton picture where Q denotes the exciton center-of-mass wave vector. After optical excitation of continuum states with the pump pulse, the sample is probed at the $A_{1s}$ (left) and $B_{1s}$ (right) exciton resonance energy. The measured signal exhibits contributions originating from instantaneous coherent polarizations as well as incoherent exciton densities, which are formed in the continuum and relax down to the energetically lowest 1s states (relaxation rate $\Gamma_{\nu+1\to\nu}$). Finally, the exciton density associated with the lowest 1s states decays slowly with a rate $\Gamma_{decay}$. **b** Calculated $\Delta R/R$ signal at the $A_{1s}$ (left) and $B_{1s}$ (right) exciton resonance energy for varying pump photon energies. All the calculated traces are normalized to the maximum value. The insets report the pump photon energy dependence of $\tau_{rise}$ of the calculated $\Delta R/R$ traces. The calculated $\tau_{rise}$ values are estimated by fitting the traces with a rising exponential convoluted with a Gaussian profile accounting for the finite temporal duration of the pump and probe pulses used in the calculation (i.e., respectively 20 fs and 15 fs). The B exciton traces exhibit, as expected, a slightly shorter rise time. This difference between the timescales is well below the temporal resolution of the pump-probe experiment and cannot be resolved by the measurements. Similar experimental $\tau_{rise}$ for the $A_{1s}$ and $B_{1s}$ excitonic resonances might also be due to the mixing of excitonic states due to intra and intervalley exchange coupling mechanism[22].

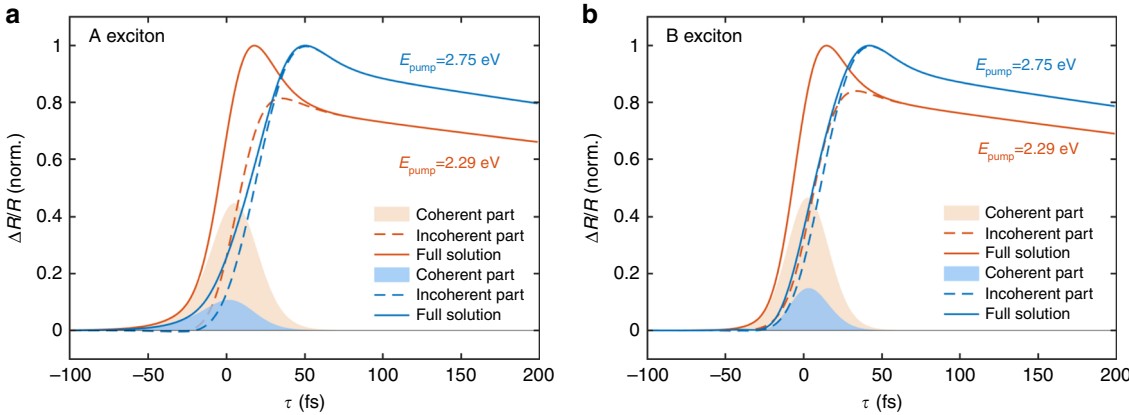

**Fig. 4 Coherent and incoherent exciton dynamics. a** Calculated $\Delta R/R$ dynamics at the energy of the $A_{1s}$ resonance considering only the coherent exciton contribution (shaded areas), only the incoherent exciton contribution (dashed lines) and both the contributions (continuous lines) for low (orange traces) and high (blue traces) pump photon energy, i.e., 2.29 and 2.75 eV, respectively. **b** The same for the dynamics at the energy of the $B_{1s}$ resonance.

created upon photoexcitation by simultaneous emission of phonons, while in the bimolecular process, excitons are created from thermalized electron-hole pairs. The observed sub-100 fs build-up dynamics suggests that the geminate mechanism mediated by strong exciton–phonon scattering is the dominant process responsible for the formation of excitons. This conclusion is further supported by the pump fluence dependent measurements (see Supplementary Fig. 8), where no change of $\tau_{rise}$ is observed for different densities of photoexcited excitons, contrary to what expected for a bimolecular formation process.

In summary, we have studied the exciton formation process in 1L-MoS$_2$ by measuring its transient optical response upon excitation with energy tunable sub-30-fs laser pulses. We resolve extremely fast and pump photon energy-dependent build-up dynamics of the $\Delta R/R$ signal around the A$_{1s}$ and B$_{1s}$ excitonic transitions. Microscopic calculations based on the TMD Bloch equations quantitatively reproduce the experimental results and explain the delayed formation of the transient signal as a result of a phonon-induced cascade-like relaxation process of high-energy incoherent excitons down to the excitonic ground state. These results shed light on the poorly explored mechanism of the exciton formation in 2D semiconductors, redefining the timescale of this process and are extremely important in view of optoelectronic applications of these materials.

## Methods

**Sample preparation.** The large area 1L-MoS$_2$ sample was grown by chemical vapor deposition on a SiO$_2$/Si substrate. The SiO$_2$ layer thickness is 300 nm. The growth procedure was carried on in a dual-zone tube furnace filled by sulfur and MoO$_3$ precursors. Further details on the growth process are reported in ref. [3]. Contrast reflectivity and photoluminescence measurements have been carried out to characterize the static optical response of the sample (see Supplementary Fig. 1).

**Pump-probe setup.** The ultrafast pump-probe experiments were carried out using a regeneratively amplified Ti:Sapphire system (Coherent Libra II), emitting 100 fs pulses centered at 1.55 eV at 2 kHz repetition rate with 4W average power. The laser drives two home-made Non-collinear Optical Parametric Amplifiers (NOPAs), which serve, respectively, as the pump and the probe. The first NOPA (probe beam) is pumped at 3.1 eV by the second harmonic of the laser and is seeded by a white-light continuum (WLC), generated in a 1-mm thick sapphire plate. The seed is amplified in a 1-mm thick beta-barium borate (BBO) crystal and compressed to nearly transform-limited duration (i.e., 7 fs) by a pair of custom-designed chirped mirrors. The probe spectrum extends from 1.75 to 2.4 eV, covering the A/B excitonic resonances of 1L-MoS$_2$. The second NOPA, used to produce the pump beam, is also pumped at 3.1 eV and uses a BBO crystal. It can be configured to amplify either the visible or the near-infrared parts of the WLC. In the former case one obtains pulses tunable from 2 to 2.5 eV, compressed to sub-20-fs duration by chirped mirrors. In the latter case one obtains pulses tunable between 1.1 and 1.4 eV, which are compressed to a nearly transform-limited sub-15-fs duration by a pair of fused silica prisms. These pulses are then frequency-doubled in a 50$\mu$m-thick BBO crystal to obtain pulses tunable between 2.29 and 2.75 eV. The overall temporal resolution of the setup is characterized via Cross-Frequency Resolved Optical Gating (X-FROG), as extensively explained in Supplementary Note 5. Pump and probe pulses are temporally synchronized by a motorized translation stage, and non-collinearly focused on the sample by a spherical mirror, resulting in spot-size diameters of ~100 μm and ~70 μm, respectively. After the interaction with the pump-excited sample, the probe pulse is spectrally dispersed on a Silicon CCD camera with 532 pixels working at the same repetition rate as the laser. The pump beam was modulated at 1 kHz by a mechanical chopper. The detection sensitivity is on the order of $10^4$–$10^5$, with an integration time of 2s. The fluence is ~5 μJcm$^{-2}$ for different pump photon energies (i.e., well below the fluence threshold for the Mott transition measured for 1L-TMDs[38,39]). No pump fluence dependence of the build-up and the relaxation dynamics was observed in the pump fluence range between 1 and 20 μJcm$^{-2}$ (see Supplementary Fig. 8). In all the experiments, pump and probe beams have parallel linear polarizations. No pump-probe polarization dependence of the build-up dynamics was detected (see Supplementary Fig. 7).

**Theory.** To get insight into the static optical properties properties of 1L-MoS$_2$ on SiO$_2$ substrate, we first solve the Wannier equation, obtaining a set of bound and continuum exciton wavefunctions $\varphi_{\lambda,q}$ with energies $\epsilon_\lambda$, cf. Supplementary Note 8. This allows to work in a convenient excitonic basis including coherent exciton polarizations $P_\lambda$, biexcitons and exciton–exciton scattering states $B_\eta$, as well as

incoherent exciton populations $N_{\lambda_1,\lambda_2,q}$. The optical response is determined by the TMD Bloch equations for the exciton amplitude $P_\lambda$ where $\lambda$ as a compound index includes the exciton state with bound and continuum excitonic states and the valley and spins of involved electrons and holes[31,40]:

$$
\begin{aligned}
\left(\partial_t + \gamma_{\lambda_1} - \frac{i}{\hbar}\epsilon_{\lambda_1}\right)P_{\lambda_1} =& -\frac{i}{\hbar}\, d_{\lambda_1}\mathcal{E}^*_{\sigma_\pm}(t) \\
&+\frac{i}{\hbar}\sum_{\lambda_2,\lambda_3,q}\hat{d}_{\lambda_1,\lambda_2,\lambda_3,q}\mathcal{E}^*_{\sigma_\pm}(t)\left(\delta_{q,0}P_{\lambda_2}P^*_{\lambda_3} + N_{\lambda_2,\lambda_3,q}\right) \\
&+\frac{i}{\hbar}\sum_{\lambda_2,\lambda_3,\lambda_4}\hat{V}_{\lambda_1,\lambda_2,\lambda_3,\lambda_4}\,P_{\lambda_2}P_{\lambda_3}P^*_{\lambda_4} \\
&+\frac{i}{\hbar}\sum_{\lambda_2,\eta}W_{\lambda_1,\lambda_2,\eta}\,B_\eta P^*_{\lambda_2}.
\end{aligned}
\tag{2}
$$

The left-hand side of Eq. (2) describes the oscillation of the exciton transition damped by a dephasing rate $\gamma_{\lambda_1}$[41–43]. The first term on the right-hand side represents the optical pump generating excitons with zero center-of-mass motion at the corners of the hexagonal Brillouin zone with $\sigma_+$ ($\xi = K$) or $\sigma_-$ ($\xi = K'$) circularly polarized light. Here, $d_{\lambda_1}$ is the optical transition matrix element and $\mathcal{E}_{\sigma_\pm}(t)$ the light field at the position of the monolayer. The latter is obtained by solving Maxwell's wave equation, which determines the reflected light[44,45]. The next contribution characterizes Pauli blocking by coherent excitons ($\sim|P|^2$) and incoherent densities ($\sim N$). The third term on the right-hand side of Eq. (2) characterizes nonlinear exciton–exciton interactions on a Hartree–Fock level ($\sim P|P|^2$), which couples bound and continuum excitonic states. Finally, the last line of Eq. (2) describes the coupling of excitons to biexcitons $B_{\eta=b}$, and two-exciton scattering continua $B_{\eta\neq b}$[46]. Here, the index $\eta$ serves as a compound index and includes the high-symmetry points and spins of the two electrons and two holes. All matrix elements are defined in ref. [18].

In order to decrease the complexity of the problem for the description of the experiment, the excitonic occupations ($N$) are treated with effective occupation numbers $\lambda_2$, which average over the details of the complex momentum distribution of incoherent exciton occupations: $\sum_{\lambda_2,\lambda_3,q}\hat{d}_{\lambda_1,\lambda_2,\lambda_3,q}N_{\lambda_2,\lambda_3,q} \approx \sum_{\lambda_2}\tilde{d}_{\lambda_1,\lambda_2}N_{\lambda_2}$ with $\tilde{d}_{\lambda_1,\lambda_2} = \hat{d}_{\lambda_1,\lambda_2,\lambda_2,0}/2$. However, the bleaching cross sections are explicitly evaluated as a function of the exciton state number $\lambda_{1/2}$. These cross sections exhibit a drastic decrease with increasing state number. The equations of motion of the exciton densities are given by:

$$
\begin{aligned}
(\partial_t + \Gamma_{decay})N_{A_{1s}/B_{1s}} =& 2\gamma_{A_{1s}/B_{1s}}\left|P_{A_{1s}/B_{1s}}\right|^2 \\
&+ \Gamma_{A_{2s}/B_{2s}\rightarrow A_{1s}/B_{1s}}N_{A_{1s}/B_{1s}},
\end{aligned}
\tag{3}
$$

$$
(\partial_t + \Gamma_{\lambda\rightarrow\lambda-1})N_\lambda = 2\gamma_\lambda|P_\lambda|^2 + \Gamma_{\lambda+1\rightarrow\lambda}N_{\lambda+1}.
\tag{4}
$$

Equations (3) and (4) describe the dynamics of the incoherent exciton densities associated with the energetically lowest A$_{1s}$/B$_{1s}$ states $N_{A_{1s}/B_{1s}}$ and higher states $N_\lambda$, respectively. $\Gamma_{decay} = 1$ meV/$\hbar$ characterizes the relaxation of the incoherent exciton densities into the ground state. Its value is adjusted to the experimental results. The first contributions to the right-hand sides of Eqs. (3) and (4) represent the formation of incoherent exciton densities out of optically excited coherent excitations by exciton–phonon scattering[47]. The terms proportional to $\Gamma_{\lambda+1\rightarrow\lambda}$ characterize the phonon-mediated relaxation from energetically higher densities with effective rates $\Gamma_{\lambda+1\rightarrow\lambda} = \frac{1\,\text{eV}}{50\,\text{fs}}\frac{1}{\epsilon_{\lambda+1}-\epsilon_\lambda}$ adapted to density functional theory calculations[48,49]. The solution of the Schrödinger equation for two electrons and two holes accesses biexcitons as well as exciton–exciton scattering continua[18,50,51]. The Heisenberg equation of motion for bound biexcitons as well as continuous exciton–exciton scattering states $B_\eta$ characterize damped ($\gamma_{\lambda_1} + \gamma_{\lambda_2}$) oscillations (energy $\epsilon_{xx,\eta}$), which are driven by two excitons $P_{\lambda_1}P_{\lambda_2}$ mediated by Coulomb interactions ($\hat{W}_{\eta,\lambda_1,\lambda_2}$):

$$
\left(\hbar\partial_t + \gamma_{\lambda_1} + \gamma_{\lambda_2} + i\epsilon_{xx,\eta}\right)B_\eta = i\sum_{\lambda_1,\lambda_2}\hat{W}_{\eta,\lambda_1,\lambda_2}P_{\lambda_1}P_{\lambda_2}.
\tag{5}
$$

Solving the set of equations of motion, Eqs. (2)–(5), together with Maxwell's wave equation[44,45] gives access to the differential reflection signal. Our simulations include coherent exciton binding energy renormalizations and other many-body effects related to the transient variation of the Coulomb screening due to optically excited A$_{1s}$ and B$_{1s}$ excitons. For the incoherent contributions we restricted our model to the Pauli blocking effect of reduced complexity whereas incoherent Coulomb renormalizations in the TMD Bloch equations, originating from incoherent exciton populations, which are formed on a similar timescale were neglected. The agreement between the calculated and measured dynamics suggests that the out-of-equilibrium optical response of 1L-MoS$_2$ on a sub-100 fs timescale is well captured by the incoherent dynamics to phase-space filling.

## Data availability

The data that support the plots within this paper and other findings of this study are available from the corresponding author upon reasonable request.

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

## Acknowledgements

We thank Dominik Christiansen for many stimulating discussions. We gratefully acknowledge funding from the Deutsche Forschungsgemeinschaft via the Projects No. 420760124 (KN 427/11-1, F.K., A.K.) as well as No. 182087777-SFB 951 (B12, M.S., A.K.). We also acknowledge support of the European Union's Horizon 2020 research and innovation program under Grant Agreement No. 734690 (SONAR, A.K. and F.S.). This project has received funding from the European Union's Horizon 2020 research and innovation program under the Marie Skodowska-Curie (grant agreement no. 734690) and from the European Research Council (ERC, grant agreement no. 816313- F.S. and no. 850875- I.K.). G.C. and S.D.C. acknowledge support by the European Union Horizon 2020 Programme under Grant Agreement No. 881603 Graphene Core 3. S.D.C. and C.T. acknowledge financial support from MIUR through the PRIN 2017 Programme (Prot. 20172H2SC4). P.J.S. and K.Y. acknowledge support from Programmable Quantum Materials, an Energy Frontier Research Center funded by the U.S. Department of Energy (DOE), Office of Science, Basic Energy Sciences (BES), under award DE-SC0019443. Work at the Molecular Foundry was supported by the Office of Science, Office of Basic Energy Sciences, of the U.S. Department of Energy under Contract No. DE-AC02-05-CH11231. This research was supported in part by the U.S. Department of Energy, Office of Science, Office of Basic Energy Sciences, Materials Sciences and Engineering Division under Contract No. DE-AC02-05-CH11231, within the van der Waals Heterostructures Program (KCWF16), which provided for sample growth, and under the sp2-bonded Materials Program (KC2207), which provided for SEM sample characterization (A.Z. and A.Y.). Support was also provided by the U.S. National Science Foundation under Grant No. DMR-1807233 which provided for additional TEM sample characterization (A.Z. and A.Y.).

## Author contributions

N.J.B., S.D.C., G.C., and P.J.S. conceived the experiment. C.T. and S.D.C. designed the experiment and performed the measurements. R.B.V., I.K., and F.S. contributed to the experimental work. F.K., M.S., and A.K. performed the theoretical calculations. K.Y., A.Y., and A.Z. prepared and characterized the sample. All the authors wrote the manuscript.

## Competing interests

The authors declare no competing interests.
