## [Peer Review File · Nature Communications]

REVIEWER COMMENTS

Reviewer #1 (Remarks to the Author):

Employing femtosecond laser pump-probe, the authors report sub-30 fs formation dynamics of excitons in monolayer MoS₂ at room temperature. Their experimental results have reached the shortest timescale so far for 2D excitonic dynamics and provided further insight of poorly understood exciton population built-up processes. Overall, I think their results are novel and convincing, which deserve to be published on Nature Communications. However, the current manuscript needs a major revision. My concerns are listed as follow:

1. The authors' successful observations crucially rely on the precise measurement of cross-correlation time of pump-probe pulses, which is about 30 fs (temporal resolution). However, it is difficult for common readers to understand why a 10 fs formation time of excitons (2.29 eV pump) can be reasonably extracted. I suggest the authors should make a brief explanation of their trick for Fig 2(d) in the maintext instead of a nontrivial fitting function in the supporting info.

2. The interpretations of Fig 2 results are incomplete. In Fig 2(c), B exciton results when pump photon energy is 2.29 eV, I think there is a negative dip before 0 fs in the rising part of transient absorption signal. Since this work is focusing on the ultrafast processes of exciton formations, the authors should not overlook such a striking feature of their results. I think the authors need to seriously clarify the observed negative dip. Where does it come from and does it influence your theoretical models? Look, the simulation for B-exciton in Fig 3(b) when pump photon energy is 2.29 eV shows a bi-exponential decay, which doesn't agree well with experimental results in Fig 2(c).

3. The presented theoretical models in Fig 3 (optical phonons involved cascade scattering processes) can partially explain why rising time linearly increases as they increase the pump photon energy as shown in Fig 2(d). However, under this model, formation time of B-exciton is expected to be shorter than that of A-exciton because there would be less intermediate scattering states for B-exciton. From the experimental results in Fig 2(d), it seems like that under the same pump conditions A and B excitons show equal formation time. But if one carefully looks at the simulation results in Fig 3(b), it seems like the rising times of B-exciton formation processes are indeed shorter than that of A-exciton. The authors should compare B-exciton formation times with A-exciton formation times as well. Please plot the simulated formation time of A/B exciton in Fig 3(b) as a function of pump photon energy and compare them with Fig 2(d).

4. To meet the high quality standards of NC, the layout and quality of all Figures in this manuscript need to be further improved. Be concise and accurate. Here I just list suggestions for some of them:

The Fig 1 fails to support the main scientific findings of this work. Is it supposed to be a phonon-mediated cascade process?

The Fig 2: the caption doesn't match with the figure content. Where is Fig 2(e)?

In Fig 3(a), arrows of pump and probe overlap and it is hard for one to distinguish. The initial and final states for Probe are OK. But for Pump, I think the initial and final states (starting and ending positions of the probe arrow) of electron transition should be at non-zero momentum instead of $k=0$ (or you call it $Q = 0$). According to the authors' drawing, the carriers relax vertically along the energy axis, the momentum conservation rule is perfectly satisfied and then optical phonons scattering will not be necessary?

5. There are some typos in the current manuscript:

On page 5, in the sentence after ref [18], "1S exciton" should be "1s exciton".

Between number and unit, there should be a space. For example: in caption of Fig 2(a), "2.29eV" should be changed to "2.29 eV". This kind of typo is very typical through the whole manuscript and please check carefully.

In summary, I think this work deserves to be published on NC, but the manuscript still needs a further review after the authors address all my above concerns.

Reviewer #2 (Remarks to the Author):

The authors report on sub-30fs resolution differential reflectivity measurements on monolayer MoS₂. They observe a delayed rise that depends linearly on the pump photon energy from 2.245 – 2.87 eV, and assign it to the exciton formation process. They then simulate this process with Bloch equations to support their assignment.

This is an important result and redefines the exciton formation time in TMDs to be approximately an order of magnitude faster than previous estimates in Ref26 and Ref30. It seems appropriate for the readership of Nat. Communications.

That said, I would like a few points clarified prior to publication

Questions/ Comments

1. The authors show experimental results that they interpret as a linear energy dependence of the exciton formation time in Fig 2.

a. Please show that the Simulations conducted and summaries in Fig 3 also reproduce this linear dependence.

b. It may be worth noting that a possibly related linear signal was reported in Ref 26 for MoSe₂ & WSe₂ and later in Cunningham et al. ACS Nano v11 p12601 '17 for WS₂, though its not clear how to compare these very different signals.

c. The authors prior published worked, Trovatiello et al. EPJ Web of Conf. v205 p05013 '19, show that this linear dependence saturates for photon energies higher than 2.7eV, which is conveniently where the measurements in the current work stop.

i. Please comment in the manuscript on this saturation, what gives rise to it, and why it is being ignored here.

ii. Please show if the current model & simulations can reproduce that saturation or whether additional physics need to be included.

d. Please add error bars to the x-axis of Figure 2D to represent the considerable bandwidth that is excited by the short pump pulses.

2. The excitation fluence is reported as 50uJ/cm².

a. Is this the incident or absorbed fluence?

b. As the pump photon energy is changed, is the incident fluence held constant or the absorbed fluence? – Since a 5x increase in absorbance occurs over this pump photon range, leading to a 4x increase in photon density, it would be important to maintain a constant absorbed fluence.

c. This fluence corresponds, for 2.29eV, to ~ 1.37E14 photons/cm² incident and ~1.49E13 photons/cm² absorbed. This exceeds both theoretical and experimental estimates of the Mott density of ~ 1E13 photons/cm² from Steinhoff et al. Nat Commun. V8 p1166 '17, and Cunningham et al. ACS Nano v11 p12601 '17 respectively. The value estimated in Methods Ref 1 of ~1E14 photons/cm² is described by the authors of that work as an overestimate. As such, this

may effect the exciton formation time due to the presence of screened Coulomb attraction.

d. Please comment on how the initial excitation density affects the simulations based on the Bloch equations. That is, are the effects of screening, shown to be very important in 2D TMDs, ignored?

e. To better justify any lack of fluence dependence within the range of absorbed fluences used here, which spans a factor of 50x:

i. Please include the 50 $\mu\text{J}/\text{cm}^2$ in data in Fig S7

ii. Please add a figure with the low and high fluence data plotted on top of one another to clearly demonstrate a lack of change in the rise time, which is currently difficult to judge.

iii. Please indicate the pump photon energy used.

3. The authors show in Fig 4 that the fast decay clearly observed for low photon energies arises from the coherent contributions to the differential reflectivity.

a. If these are coherent contributions, wouldn't an experiment examining the degree of linear polarization, or degree of circular (valley) polarization, better support this claim?

b. In Fig S6 shows that this fast decay is only observed for cross-polarized pump and probe. This is opposite the expectation for a coherent process in 2D TMDs. Please explain.

4. Much of the validity of the analysis hinges upon interpreting the changes in reflected light intensity as due to Pauli Blocking. While the authors support this claim with comparisons to integrated spectra in Fig S2, there are marked differences in prior published works that require some comment and explanation.

a. The authors of this work have a prior publication on MoS₂, Pogna ACS Nano v10 p1182 '16, demonstrating that Pauli Blocking cannot explain the transient absorption spectra at a delay of 300fs after excitation. However the results here show no spectral shifts until after 1ps. Please explain why such different results are reported here.

b. Similarly, the differential reflectivity here, e.g. in Fig S3, differ significantly from past spectra: e.g. Sim et al. PRB v8 p075434 '13 and Borzda et al. Adv Funct Mater v25 p3351 (co-authors by some of the current authors), both of which report numerous positive and negative features which are either assigned to spectral shifts or charged excitons. Please explain why such different results are reported here.

Reviewer #1 (Remarks to the Author):

Employing femtosecond laser pump-probe, the authors report sub-30 fs formation dynamics of excitons in monolayer MoS₂ at room temperature. Their experimental results have reached the shortest timescale so far for 2D excitonic dynamics and provided further insight of poorly understood exciton population built-up processes. Overall, I think their results are novel and convincing, which deserve to be published on Nature Communications. However, the current manuscript needs a major revision. My concerns are listed as follow:

Reply: We thank the Reviewer for judging our results “novel and convincing”, and for recommending publication of the paper in Nature Communications. In the following we fully address the Reviewer’s concerns.

1. The authors' successful observations crucially rely on the precise measurement of cross-correlation time of pump-probe pulses, which is about 30 fs (temporal resolution). However, it is difficult for common readers to understand why a 10 fs formation time of excitons (2.29 eV pump) can be reasonably extracted. I suggest the authors should make a brief explanation of their trick for Fig 2(d) in the main text instead of a nontrivial fitting function in the supporting info.

Reply: We agree with the Referee that the part of the manuscript related to the experimental estimation of the exciton formation time has to be discussed more thoroughly, since it is the core of this work. For this reason, we moved the technical section describing the fitting of the differential reflectivity ($\Delta R/R$) time traces to the main text. We stress here that the measurement of such a fast process relies on the employment of few-optical-cycle laser pulses, on their careful temporal characterization by nonlinear optical effects, and on the deconvolution of the data and the pulse characteristics. The rising edge of the $\Delta R/R$ traces is fitted by an exponential build-up convoluted with a Gaussian profile, which is the instrumental response function (IRF) and corresponds to the experimentally measured cross-correlation trace between pump and probe pulses. In the Supplementary Information, we show that the rising edge of the signal is not well reproduced by the convolution between the IRF and a Heaviside function (which would correspond to an instantaneous response). Only after including a finite exponential rise time in the fitting function, we can reproduce satisfactorily the experimental formation time (see Fig. S6, where we show identical fitting functions with and without the additional rise time). For the lower pump photon energy (i.e. 2.29 eV, left panel in Fig. S6), a 10-fs build-up must be included in order to properly fit the data and to reproduce the measured $\Delta R/R$ dynamics. This value, although slightly smaller than the width of the IRF and affected by large error bars, can still be reliably extracted thanks to the high signal to noise ratio of our experiment. Of course, the estimation of the rise time for the higher pump photon energies (corresponding to a slower exciton formation) is affected by a smaller uncertainty.

Action taken:

We moved part of the Supplementary Information section related to the fitting procedure to the main text and we added the following sentences:

“The build-up times are estimated by fitting the time traces, in the temporal window between -100 fs and 200 fs, to the function:

$$f(t) = \left(1 - e^{-\frac{t}{\tau_{rise}}}\right) * H(t) * \left(A_1 e^{-\frac{t}{\tau_1}} + A_2 e^{-\frac{t}{\tau_2}}\right)$$

where $H(t)$ is the Heaviside function, τ_1 and τ_2 are decay time constants and A_1 and A_2 are the amplitudes of each decay component. The fitting function is convoluted with a Gaussian profile corresponding to the instrumental response function, which is the experimentally measured cross-correlation profile between pump and probe (see Methods and Supplementary Information for the temporal characterization of the pulses). We stress that only after including a finite exponential rise time in the fitting function, we can reproduce satisfactorily the experimental formation time (see Fig. S6).”.

We also modified the fitting procedure part of the Supplementary Information as follows:

“As reported in the main text, the $\Delta R/R$ time traces are fitted by a rising exponential, describing the build-up of the transient signal, and two exponential decays related to the different timescales of the exciton relaxation processes. This fitting function is convoluted with a Gaussian function accounting for the finite instrumental response function (IRF). The results of the fit are reported in Fig. S6 for the traces measured with the lowest and highest pump photon energies. The IRF is carefully determined by cross-correlation frequency resolved optical gating (X-FROG) measurements (see Section 5). The dashed lines in Fig. S6 are the fitting functions without the contribution of a rising exponential, i.e. assuming an instantaneous response. It is clear that only by including the effect of a finite rise time in the fitting function, the $\Delta R/R$ traces at the early femtosecond timescale can be satisfactorily reproduced.”.

2. The interpretations of Fig 2 results are incomplete. In Fig 2(c), B exciton results when pump photon energy is 2.29 eV, I think there is a negative dip before 0 fs in the rising part of transient absorption signal. Since this work is focusing on the ultrafast processes of exciton formations, the authors should not overlook such a striking feature of their results. I think the authors need to seriously clarify the observed negative dip. Where does it come from and does it influence your theoretical models?

Reply: We thank the Reviewer for this insightful comment. We attribute the weak negative dip observed for negative times around the B exciton signal at 2.29 eV excitation photon energy to the so called pump-perturbed free induction decay (PPFID) (see Refs. C. H. Brito Cruz et al., IEEE Journal of Quantum Electronics, **24**, 2, 1988 and L. Luer et al., ACS Nano, **4**, 4265, 2010). In this process, the coherent interaction between the perturbed free-induction decay signal (i.e. the interaction between the pump and the probe-induced polarization) and the probe pulse, leads to a periodic modulation of the transmission of the probe pulse at negative time delays.

By including in the calculation only the coherent coupling and the PPFID effect, the $\Delta R/R$ traces display a transient signal at negative delays changing sign from negative to positive at increasing photon energies around the excitonic resonances, as shown in Fig. R1. This change of sign is less pronounced around the A excitonic resonance because of the higher energy difference with respect to the pump excitation.

Fig. R1: Calculated effect of the pump-perturbed free induction decay on the transient exciton dynamics of 1L-MoS₂, following a pump excitation at 2.29 eV.

Similar features appear in the experimental $\Delta R/R$ maps before the pump arrival. Figure R2 reports the transient map of Fig. 2a of the main manuscript, here drawn on a different color scale in order to better distinguish positive (red) from negative (blue) transient signals. For each excitonic resonance there is a fast negative dip just before time zero at slightly lower photon energies than the peak of the exciton bleaching signal. A weaker

negative signal at negative delays appears at the same energy of the exciton bleaching only for the B exciton and for the lowest pump photon energy.

Fig. R2: left panel: $\Delta R/R$ map measured across A and B excitonic transition of 1L-MoS₂, following an excitation photon energy of 2.29 eV; right panel: zoom around the B exciton resonance.

By including also population effect in the calculated $\Delta R/R$ trace reported in Fig. 3 of the manuscript, the PPFID signals are almost obscured due to contributions from higher bound and continuous exciton states. We attribute the underestimation of this effect to disorder, sample imperfections, and possibly a doping density of electrons.

Anyhow, we stress that the intensity of the signal at negative delay is much lower (i.e. it is only a few percent) than the bleaching signal which is the main object of our investigation. Since these signals occur at negative delays, they do not affect the build-up dynamics and do not prevent a precise estimation of the exciton formation time.

Action taken: We have added to the manuscript the following paragraph: “The weak negative dip observed for negative times in the B exciton signal for 2.29 eV excitation photon energy is attributed to the pump-perturbed free induction decay (PPFID) [C. H. Brito Cruz et al., *IEEE Journal of Quantum Electronics*, 24, 2, 1988]. In this process the free-induction decay field emitted by the sample excited by the probe pulse is perturbed by the interaction with the pump pulse. The PPFID signal does not prevent a precise estimation of the exciton formation time because it occurs before the rising edge of the transient signal and its amplitude is more than one order of magnitude lower than the peak of the excitonic bleaching.”.

Look, the simulation for B-exciton in Fig 3(b) when pump photon energy is 2.29 eV shows a bi-exponential decay, which doesn't agree well with experimental results in Fig 2(c).

Reply: We thank the Referee for noticing this discrepancy. Indeed, the calculated decay dynamics of the B excitonic trace for pump energy at 2.29 eV, display a somewhat different behavior than the corresponding experimental one. Our theoretical model captures very well the relaxation cascade process of high-energy excitons down to their incoherent ground state, while the recombination dynamics (radiative and non-radiative) of the lower energy incoherent excitons (i.e. A_{1s} and B_{1s}) is regulated by a single phenomenological parameter (i.e. Γ_{decav}). We believe this discrepancy tells us that a single parameter is not enough to capture the multiple relaxation mechanisms that determine the decay dynamics of the ground state incoherent excitons and a more refined model is required, which, however, is out of the scope of the present paper.

3. The presented theoretical models in Fig 3 (optical phonons involved cascade scattering processes) can partially explain why rising time linearly increases as they increase the pump photon energy as shown in Fig 2(d). However, under this model, formation time of B-exciton is expected to be shorter than that of A-exciton because there would be less intermediate scattering states for B-exciton. From the experimental results in Fig 2(d), it seems like that under the same pump conditions A and B excitons show equal formation time. But if one carefully looks at the simulation results in Fig 3(b), it seems like the rising times of B-exciton formation processes are indeed shorter than that of A-exciton. The authors should compare B-exciton formation times with A-exciton formation times as well. Please plot the simulated formation time of A/B exciton in Fig 3(b) as a function of pump photon energy and compare them with Fig 2(d).

Reply: We agree with the Referee that, within our experimental time resolution, we indeed observe a very similar rise time of A and B excitons. The Referee is also right in stating that in our numerical simulations the build-up of B excitons is actually shorter than that of the A excitons, in line with our proposed formation process. As the Referee correctly states, this numerical result is due to the phonon mediated relaxation which is delayed for energetically lower excitons. This naturally implies a shorter expected formation time of B excitons and we see no reason why the opposite should be expected. However, as the measured and simulated build-up times are extremely fast, we believe that the temporal resolution of our experiments is not sufficient to capture this small difference of 5-7 fs. Following the Reviewer's suggestion, we added to the inset of Fig. 3b the plot of the rise time of the calculated A/B excitonic traces in order to have a more direct comparison with the rise times (reported in Fig. 2d) which are extracted from the experimental traces in Fig. 2c. As we reported in the inset, the calculated rise time for the A exciton is longer with respect to that of the B excitons. Given the temporal resolution and the error bars of our measurements, however, we feel that we cannot confidently resolve this difference in our experiments. Additionally, one should consider that intra- and inter-valley exchange coupling mixes A and B excitonic states (see M. Selig et al. Physical Review Research 1, 022007(R) (2019)). This coupling might result in a similar build up dynamics measured across the two excitonic resonances.

Action taken: We added a plot of the calculated rise time of the A/B excitonic traces vs. the pump photon energy as insets in Fig. 3b, here reported as Fig. R3. The following has been added to the manuscript:

"The insets report the rise time extracted for the calculated A and B excitonic traces. The B exciton traces exhibit, as expected from the model, a shorter rise time. Given the temporal resolution and the error bars of our measurements, however, this difference cannot be experimentally resolved. Similar experimental τ_{rise} for the A_{1s} and B_{1s} excitonic resonances might also be due to the mixing of excitonic states due to intra and intervalley exchange coupling mechanism[23]."

Fig. R3: Simulation of the exciton formation process. a) Schematic illustration of the relaxation model in the exciton picture where Q denotes the exciton center-of-mass wave vector. After optical excitation of continuum states with the pump pulse the sample is probed at the A_{1s} (left) and B_{1s} (right) exciton resonance energy. The measured signal exhibits contributions originating from instantaneous coherent polarizations as well as incoherent exciton densities which are formed in the continuum and relax down to the energetically lowest $1s$ states (relaxation rate $\Gamma_{\nu+1 \rightarrow \nu}$). Finally, the exciton density associated with the lowest $1s$ states decays slowly with a rate Γ_{decay} . b) Calculated Pauli blocking contributions to the pump-probe signals at the A_{1s} (left) and B_{1s} (right) exciton resonance energy for varying pump photon energies. All the calculated traces are normalized to the maximum value. The insets report the pump photon energy dependence of τ_{rise} of the calculated bleaching signals for A_{1s} and B_{1s} excitonic resonances. The calculated τ_{rise} are estimated by fitting

the traces with a rising exponential convoluted with a Gaussian profile accounting for the finite temporal duration of the pump and probe pulses used in the calculation (i.e. respectively 20 fs and 15 fs). The B exciton traces exhibit, as expected, a slightly shorter rise time. This difference between the time scales is well below the temporal resolution of the pump-probe experiment and cannot be resolved by the measurements.”.

4. To meet the high quality standards of NC, the layout and quality of all Figures in this manuscript need to be further improved. Be concise and accurate. Here I just list suggestions for some of them: The Fig 1 fails to support the main scientific findings of this work. Is it supposed to be a phonon-mediated cascade process?

Reply and action taken: following the Reviewer’s suggestions, we slightly modified Figure 1 adding the graphics of the phonon mediated cascade. The new version of the figure is reported below, as Fig. R4.

Fig. R4: Sketch of the experiment. a) Cartoon of the exciton formation process after photo-injection of free electron-hole pairs. b) Schematic illustration of the single particle band structure of 1L-MoS₂ at the K/K' points. The two arrows represent A/B excitonic transitions, split due to the strong spin-orbit interaction at the K/K' points of the Brillouin zone. c) Sketch of the pump-probe experiment. A few-optical-cycle laser pulse injects free electron/hole pairs at increasing energies above the exciton continuum (E_G). These quasiparticles lose their initial kinetic energy and scatter down, via a cascade process mediated by phonons, to lower-lying discrete excitonic states until they reach the 1s excitonic state. The timescale $\tau(E)$ of this relaxation process is determined by measuring the absorption change of a probe beam, tuned on resonance with the 1s state, due to the Pauli blocking effect.

The Fig 2: the caption doesn't match with the figure content. Where is Fig 2(e)?

Reply and action taken: We thank the Referee for spotting this mistake, which we corrected in the revised version of the manuscript.

In Fig 3(a), arrows of pump and probe overlap and it is hard for one to distinguish. The initial and final states for Probe are OK. But for Pump, I think the initial and final states (starting and ending positions of the probe arrow) of electron transition should be at non-zero momentum instead of $k=0$ (or you call it $Q = 0$). According to the authors' drawing, the carriers relax vertically along the energy axis, the momentum conservation rule is perfectly satisfied and then optical phonons scattering will not be necessary?

Reply: We agree with the Referee that our previous schematic illustration (Fig. 3a) was confusing and we agree the scattering process involves states with non-zero momentum. Every excitonic state (also in the continuum) features a parabolic dispersion representing the center-of-mass motion. The initial and final states of pump and probe are at zero exciton center-of-mass wave vector $Q=k_{\text{electron}}-k_{\text{hole}}=0$. This is due to the fact that only $Q=0$ states can be optically excited and detected, i.e., the involved electron and hole have identical wave vector $k_{\text{electron}}=k_{\text{hole}}$ (see Supplementary Note 8). The finite center-of-mass wave vector Q occurs due to phonon scattering. Please note that the optical excitation at $Q=0$ should not be confused with $k_{\text{electron}}=k_{\text{hole}}=0$. Instead, it only accounts for the negligibly small photon wave vector which implies $Q=k_{\text{electron}}-k_{\text{hole}}=0$.

Action taken: In order to clarify this misunderstanding, we modified the schematic illustration in Fig. 3a following the Referee's suggestions. We have drawn the arrows of the pump and the probe with different thickness in order to better distinguish them. We added a short note to the caption which states that Q denotes the exciton center-of-mass wave vector instead of the electron or hole wave vector. We have also redrawn the scattering arrows to make it clear that the scattering processes involves states with non-zero momentum. The new Fig. 3 is here reported as Fig. R3.

5. There are some typos in the current manuscript:

On page 5, in the sentence after ref [18], "1S exciton" should be "1s exciton".

Between number and unit, there should be a space. For example: in caption of Fig 2(a), "2.29eV" should be changed to "2.29 eV". This kind of typo is very typical through the whole manuscript and please check carefully.

Reply: we thank the Referee for the careful reading of the manuscript and for catching these typos.

Action taken: In the new version of the manuscript all these typos have been corrected.

In summary, I think this work deserves to be published on NC, but the manuscript still needs a further review after the authors address all my above concerns.

Reply: we have carefully addressed all the Reviewer's concerns and we now hope that she/he will find our paper suitable for publication in Nature Communications.

Reviewer #2 (Remarks to the Author):

The authors report on sub-30fs resolution differential reflectivity measurements on monolayer MoS₂. They observe a delayed rise that depends linearly on the pump photon energy from 2.245 – 2.87 eV, and assign it to the exciton formation process. They then simulate this process with Bloch equations to support their assignment.

This is an important result and redefines the exciton formation time in TMDs to be approximately an order of magnitude faster than previous estimates in Ref26 and Ref30. It seems appropriate for the readership of Nat. Communications.

That said, I would like a few points clarified prior to publication

Reply: we thank the Reviewer for the effective summary of our paper, for judging our study “an important result” that “redefines the exciton formation time in TMDs” and for deeming it appropriate for publication in Nature Communications. In the following we address the concerns raised by the Reviewer.

Questions / Comments

1. The authors show experimental results that they interpret as a linear energy dependence of the exciton formation time in Fig 2.

Reply: We thank the Referee for this comment, and we agree with the Reviewer that the data appears to be linear. However, we note that, due to the error bars associated with the determined rise times, it is difficult to make a clear (i.e., statistically significant) statement on the functional dependence of the rise time on the pump photon energy, even if the data in Fig. 2d could be fitted by a linear function. In the manuscript, acknowledging that error bars do not allow for an unambiguous determination of a linear dependence, we attempted to avoid asserting that the trend obeys a specific power law, and rather only state that the rise time “monotonically increases with the initial excess energy of the photoinjected carriers”.

Action taken: We have changed the sentence “(1) the formation time of the excitons increases linearly with increasing energy” into “(1) the formation time of the excitons monotonically increases with the initial excess energy of the photoinjected carriers”.

a. Please show that the Simulations conducted and summaries in Fig 3 also reproduce this linear dependence.

Reply: Following the Reviewer’s suggestion, we reported in the inset of Fig.3 (here reported as Fig. R3) plots of the fitted rise times of A and B excitons, τ_{rise} , for the calculated $\Delta R/R$ traces. As shown in the new insets, τ_{rise} of both the A and B excitons linearly increases with the pump photon energy. As expected, the rate of increase is higher for the lower-energy exciton. We stress that the small difference between the calculated rise times of the two excitonic traces (i.e. 5-7 fs) cannot be resolved in the experiment because of the limited temporal resolution and error bars of the determined time constants. Moreover, intra and intervalley exchange coupling mixes the A_{1s} and B_{1s} excitonic states (see Ref. 23). This coupling might lead to similar rise times for the build-up dynamics of the two excitons.

Action taken: We added two insets in Fig. 3b (left and right panel) reporting the evolution of the calculated τ_{rise} of A and B excitons at increasing pump photon energies. The following sentence has been added to the caption of Fig.3: “*The insets report the pump photon energy dependence of τ_{rise} of the calculated bleaching signals for A_{1s} and B_{1s} excitonic resonances. The calculated τ_{rise} are estimated by fitting the traces with a rising exponential convoluted with a Gaussian profile accounting for the finite temporal duration of the pump and probe pulses used in the calculation (i.e. respectively 20 fs and 15 fs).*”.

b. It may be worth noting that a possibly related linear signal was reported in Ref 26 for MoSe₂ & WSe₂ and later in Cunningham et al. ACS Nano v11 p12601 '17 for WS₂, though it's not clear how to compare these very different signals.

Reply: These are indeed interesting studies and we have added a reference to the Cunningham et al. work to our manuscript. Both articles mentioned by the Reviewer report studies of the transient optical response of single layer TMDs at different pump photon energies in a similar fashion, as we did in our manuscript. The papers focus on the relaxation dynamics of the excitonic resonances in TMDs, but do not treat their build-up dynamics because they lack the temporal resolution needed to do so. (Specifically, the temporal resolution is

a factor of ≈ 10 (i.e. 150 fs, in the paper by Cunningham et al.) and ≈ 20 (i.e. 400 fs, in Ref. 26) lower than in our work). This difference in resolution along with the fact that they are measured on different TMDs makes it difficult to compare the different dynamics, as the Reviewer notes.

Action taken: we added a citation for the paper by Cunningham et al. as Ref. [28].

c. The authors prior published worked, Trovatello et al. EPJ Web of Conf. v205 p05013 '19, show that this linear dependence saturates for photon energies higher than 2.7eV, which is conveniently where the measurements in the current work stop.

*i. Please comment in the manuscript on this saturation, what gives rise to it, and why it is being ignored here.
ii. Please show if the current model & simulations can reproduce that saturation or whether additional physics need to be included.*

Reply: We thank the Reviewer for this comment. In the conference proceeding mentioned by the Reviewer, we reported the exciton build-up dynamics at pump photon energies between 2.3 eV and 2.75 eV plus a single value of the rise time measured at a much higher pump energy (i.e. around 3.75 eV). The light pulses in the lower energy range are generated by compressing and frequency doubling the tunable output of a near-IR optical parametric amplifier. On the other hand, the pump pulses around 3.75 eV are produced by a different nonlinear optical process (sum-frequency generation with the 1.55 eV laser pulse), allowing us to reach the UV region, but the pulses cannot be easily tuned around that photon energy without losing temporal resolution. After the publication of this proceeding, we tried to generate light pulses with duration below 20-30 fs in the region between 2.75 eV and 3.75 eV to close the experimental gap, but we have not yet succeeded. From the data in the proceeding, it seems that the rise time above 2.75 eV reaches a saturation but we cannot provide a precise trend of the exciton dynamics in this broad (almost 1 eV) energy window. This lack of data is the first reason why we decided not to include the experimental point at 3.75 eV in the manuscript.

Moreover, while the photoexcitations below 2.75 eV involve electron and hole states of the conduction and valence bands around the K/K' points of the Brillouin zone, the photoexcitation around 3.75 eV is in resonance with electronic states also around the Γ and M points. The A/B exciton bleaching signals observed upon this high-energy excitation are likely to be originated by scattering processes which are different from the ones mediating the exciton cascade relaxation at the K/K' points. High-momentum acoustic phonons can mediate the scattering of e/h pairs close to the Γ point towards the electronic states around the K/K' points and could be responsible for the ultrafast onset of the bleaching signals around A and B excitonic resonances. The coexistence of several scattering processes involving both optical and acoustic phonons and electronic states located in different symmetry points of the Brillouin zone tentatively explains why the exciton formation time upon excitation at 3.75 eV does not follow the pump photon energy trend observed in the manuscript. Moreover, 3.75 eV is almost two times the exciton energy. At this excitation energy, also Auger-like relaxation processes (i.e. one high energy exciton is annihilated while two A/B excitons are created) could come into play, which would lead to a faster relaxation of the excitons towards the band edge.

Regarding the simulations, the exciton band structure and the underlying dynamics for low pump photon energies in TMDs are well captured by treating the high-symmetry points in an effective mass approximation (see Ref. 2D Mater. **2**, 022001 (2015)). However, this approximation breaks down for excitation far above the band gap, which is the case for energies above ≈ 3 eV. Here, the band structure is too complicated for a parabolic approximation, which we use in the theoretical description. Therefore, as we aim at a consistent experiment-theory comparison for all of the data discussed in our manuscript (which is not possible for the mentioned 3.75 eV data), we do not include these data in the main part of the manuscript.

Action taken: In order to maintain the symmetry between experimental and theoretical data in the main text, we decided to include the high pump photon energy dynamics as a new figure in the Supplementary Information section of the paper, as Fig. S3 (here reported as Fig. R5). In addition, we added the following in the main text:

"Figure 2d unambiguously shows that τ_{rise} monotonically increases with the initial excess energy of the photoinjected carriers. Interestingly, for a higher pump photon energy (i.e. 3.75 eV), we observe that τ_{rise} sensitively deviates from the increasing energy trend reported in Fig. 2d (see Supplementary Note 2, Fig. S3). We can tentatively explain this saturation of the build-up dynamics as a result of different phonon-mediated scattering processes involving electronic states far away from the K/K' points."

We therefore added the following section in the SI:

"Figure S3 reports the $\Delta R/R$ formation dynamics measured upon excitation with a pump pulse centered around 3.75 eV, i.e. almost 1 eV higher than the maximum value of the pump photon energy reported in the

manuscript. In this experiment, the UV pump pulses are obtained by frequency up-conversion of a broadband NOPA as explained in details in Ref. [R. Borrego-Varillas et al., Applied Sciences **8**, 989, 2018]. These pulses are compressed down to 20 fs by a pair of prisms and temporally characterized by a two-dimensional spectral-shearing interferometry method (2DSI) in the UV region. As shown in the figure, τ_{rise} does not follow the same trend observed up to 2.75 eV and it seems to maintain a constant value between 30-40 fs. We fit the rising edge dynamics with the same fitting function described in the main text and we obtain the following values for τ_{rise} of A_{1s} and B_{1s} excitons: 34 ± 3 fs and 37 ± 5 fs, respectively.

This result points towards a possible saturation effect of the exciton formation dynamics at higher photon energies. Unfortunately, we are not able to perform pump-probe experiments with sub-30-fs temporal resolution using pump pulses tunable in the broad energy window ranging from 2.75 eV and 3.75 eV and to give a precise estimation of the energy where this saturation effect sets in. When the laser pulse is tuned to such a high energy, optical transitions involving electronic states close to Γ and M symmetry points can be activated in addition to the optical transitions around the K/K' points.

We can tentatively explain the deviation of the exciton formation time from the monotonic increase trend observed at lower pump photon energies, as a result of a Pauli blocking effect due to the quick scattering of photoexcited carriers from different symmetry points (i.e. Γ and M) of the Brillouin zone to the K/K' points. These scattering processes are mediated by large-momentum acoustic phonons. Moreover, 3.75 eV is almost two times the exciton energy. At this excitation energy, also Auger-like relaxation processes (i.e. one high energy exciton is annihilated while two A/B excitons are created) could come into play, possibly leading to a faster relaxation of the excitons towards the band edge.”

Fig. R5: UV photon energy excitation. $\Delta R/R$ traces measured across A_{1s} (left panel) and B_{1s} (right panel) excitonic resonances for $E_{\text{pump}}=3.75$ eV (yellow lines). In both graphs we report the $\Delta R/R$ traces at $E_{\text{pump}}=2.75$ eV for comparison (dashed lines). For both the excitation conditions the two dynamics temporal traces have almost the same build-up time.

d. Please add error bars to the x-axis of Figure 2D to represent the considerable bandwidth that is excited by the short pump pulses.

Reply and action taken: following the Reviewer’s suggestion, we added error bars to the x-axis of panel d in Fig. 2 (here reported as Fig. R6) taken as the full-width at half maximum of the excitation pulse spectra.

Fig. R6: Energy dependent exciton formation process. $\Delta R/R$ maps measured on 1L-MoS₂ photo-excited a) at 2.29 eV and b) at 2.75 eV. The measurements are performed at room temperature. The horizontal dashed lines pass through the maximum of the $\Delta R/R$ spectrum at the energies of the A_{1s}/B_{1s} exciton transitions, while the vertical lines mark the temporal range from 10% to 90% of the build-up signal. The excitation fluence is $\sim 5 \mu\text{J}/\text{cm}^2$. Pump and probe beams have parallel linear polarizations. The time zero is defined, for each measurement, as the maximum of the cross correlation signal between the pump and the probe pulses as explained in the Methods. c) Temporal cuts of the $\Delta R/R$ maps measured across the A_{1s} and B_{1s} excitonic resonances for increasing pump photon energy. d) Pump photon energy dependence of τ_{rise} . Horizontal error bars are determined by the bandwidth of the pump pulses; vertical error bars are obtained from the fits of the time traces.

2. The excitation fluence is reported as $50\mu\text{J}/\text{cm}^2$.

a. Is this the incident or absorbed fluence?

Reply: We thank the Reviewer for this comment. Actually, we checked more carefully our laboratory notebook and realized that we have mistakenly calculated $50 \mu\text{J}/\text{cm}^2$. The correct value of the incident fluence is about $5 \mu\text{J}/\text{cm}^2$ (corresponding to an energy of $\sim 0.5 \text{ nJ}$ focused over a spot of $\sim 100 \mu\text{m}$) and not $50 \mu\text{J}/\text{cm}^2$, which is a quite high value (we agree with the Referee on that). We apologize for this error.

Action taken: We added the corrected value of the pump fluence in the new version of the manuscript. We have changed the sentence in the caption of Fig. 2 “The excitation fluence is $50 \mu\text{J}/\text{cm}^2$ ” into “The incident fluence is $5 \mu\text{J}/\text{cm}^2$ ”. We have changed the sentence in the Methods “The fluence is $50 \mu\text{J}/\text{cm}^2$ for different pump photon energies” into “The incident fluence is $5 \mu\text{J}/\text{cm}^2$ for different pump photon energies”.

b. As the pump photon energy is changed, is the incident fluence held constant or the absorbed fluence? – Since a 5x increase in absorbance occurs over this pump photon range, leading to a 4x increase in photon density, it would be important to maintain a constant absorbed fluence.

Reply: We thank the Reviewer for this comment. In principle the Reviewer is right for transient absorption measurements performed on samples grown on transparent substrates: for a constant incident fluence, an increase of the pump photon energy would determine a higher density of injected e/h pairs, due to increased absorbance, and therefore a higher pump-probe signal. The situation is different for samples deposited on Si/SiO₂ substrates, which is our case. Here the increase of the $\Delta R/R$ signal at higher pump photon energies is overcompensated by an interference effect due to the multiple reflections of the incoming beams in the SiO₂ layer. This effect has been observed and modelled in Ref. [E. Kim et al. Adv. Mat. Inter. **5**, 1701637, 2018]. Due to this effect, we observe a reduction of the pump-probe signal at increasing pump photon energies

starting from $E_{\text{pump}} = 2.29 \text{ eV}$. In the experiment reported in Fig. 2 the incident fluence was set to $5 \mu\text{J}/\text{cm}^2$ and slightly changed around this value in order to maintain the same intensity of the $\Delta R/R$ signals. All the $\Delta R/R$ traces in Fig. 2, are then normalized to their maximum values in order to better compare the build-up time.

As explained in the next point of the response, we have also varied the incident fluence over an order of magnitude (see Fig. S8), which more than compensates for the variation in absorbed photons from one pump energy to the other, without observing any change of the dynamics.

c. This fluence corresponds, for 2.29eV, to $\sim 1.37E14 \text{ photons}/\text{cm}^2$ incident and $\sim 1.49E13 \text{ photons}/\text{cm}^2$ absorbed. This exceeds both theoretical and experimental estimates of the Mott density of $\sim 1E13 \text{ photons}/\text{cm}^2$ from Steinhoff et al. Nat Commun. V8 p1166 '17, and Cunningham et al. ACS Nano v11 p12601 '17 respectively. The value estimated in Methods Ref 1 of $\sim 1E14 \text{ photons}/\text{cm}^2$ is described by the authors of that work as an overestimate. As such, this may affect the exciton formation time due to the presence of screened Coulomb attraction.

Reply: We apologize again for the error. For the correct value of the incident fluence, we have a photoexcited carrier density of $\sim 1.5 \times 10^{12} \text{ cm}^{-2}$, so that we are safely below the excitonic Mott transition threshold reported by Steinhoff et al. We also varied the fluence over more than one order of magnitude and we observed no change of the rise time of the bleaching signal, as shown in Fig. S8. This means that the screening of the Coulomb attraction plays a secondary role in the process of the exciton formation. We stress that, for the range of fluences used in our experiment, the intensity of the pump-probe signal is comparable or even lower than the one reported in Ref. [26] and in the paper by Cunningham et al., i.e. of the order of 10^{-3} .

Action taken: we added a citation for the paper by Steinhoff et al. as Ref. [2] of the Methods.

d. Please comment on how the initial excitation density affects the simulations based on the Bloch equations. That is, are the effects of screening, shown to be very important in 2D TMDs, ignored?

Reply: Here the Referee addresses a very important point. In general, one has to distinguish between two screening contributions: firstly, the always present dielectric screening by the environment and energetically high (not dynamically treated) bands in TMDs, and, secondly, the additionally induced screening due to a build-up of optically induced excitations. We have the impression that the Referee refers to the second contribution, but we address both points separately.

Firstly, we discuss the environment: we use a static Coulomb potential V_q accounting for finite thickness effects which is screened by a non-local dielectric function ϵ_q parametrized from DFT calculations [Scientific Reports 7, 39844 (2017)]. The screening function captures the q -dependence of the effective dielectric constant of the monolayer and the dielectrically static substrate effects. This Coulomb potential contains as limiting cases the ideal 2D Coulomb potential for vanishing semiconductor thicknesses as well as the Rytova-Keldysh potential for small q (derived originally for thin semiconductor films) and gives accurate results of the exciton binding energy [Phys. Rev. B. 88, 045318 (2013), Phys. Rev. Lett. 113, 076802 (2014)].

Secondly, we address the problem of dynamical screening which depends – as the Referee states – on the photoexcited charge carrier density built up due to optically excited quasiparticles (excitons): we start from a Hartree-Fock ground state with no electron gas or doping background. The laser pulses then build up an optically induced excitation density leading to screening or energy-renormalizations due to a local field as well as even higher bound states such as biexcitons. The effects of screening the Referee asks about occur in our description from a dynamics controlled truncation scheme [Zeitschrift für Physik B Condensed Matter 93, 195 (1994)] via density dependent renormalizations of excitons due to photoexcited charge carrier densities. We treat these renormalizations beyond the Hartree-Fock approximation by explicitly including correlation effects stemming from exciton-exciton scattering states. In our many particle hierarchy, these effects dominate the onset of screening. We find that the Hartree-Fock approximation induces an A_{1s} exciton density dependent blue shift of $9.1 \cdot 10^{-12} \text{ meV cm}^2$ whereas correlation effects lead to a red shift of $7.7 \cdot 10^{-12} \text{ meV cm}^2$. These opposite shifts largely compensate each other leading only to a small influence of energy renormalizations which is also observed in our experiments. Therefore, by including now Hartree-Fock as well as correlation effects at the same time our main findings and interpretation of the results are unchanged.

Action taken: We addressed the first point by including a discussion to Supplementary Note 7. Moreover, we addressed the second point by revising the Method's section and by including the Coulomb screening effect in the theoretical model. The new calculated $\Delta R/R$ signals are reported in Fig. 3 and 4 of the revised version of the manuscript.

e. To better justify any lack of fluence dependence within the range of absorbed fluences used here, which spans a factor of 50x:

i. Please include the 50 $\mu\text{J}/\text{cm}^2$ in data in Fig S7

Reply: we thank the Reviewer for pointing out this discrepancy. We have clarified this point above.

ii. Please add a figure with the low and high fluence data plotted on top of one another to clearly demonstrate a lack of change in the rise time, which is currently difficult to judge.

Reply and action taken: we have modified Fig. S8 accordingly, here reported as Fig. R7.

Fig. R7: Pump fluence dependence. Normalized $\Delta R/R$ dynamics (top left) and spectra (top right) measured at A_{1s}/B_{1s} excitonic resonances at different pump fluences in the range 1-20 $\mu\text{J}/\text{cm}^2$ with pump photon energy of 2.75eV. The peak of A and B exciton bleaching signal displays a linear dependence with the pump fluence in the examined range (bottom left). The transient spectra at different time delays (bottom right) show the spectral profile of A and B excitons in the temporal region of explored in the experiment.

iii. Please indicate the pump photon energy used.

Reply and action taken: we apologize for the missing pump energy value. The excitation photon energy is 2.75 eV. We have added this to the caption of Fig. S8, which now reads: **“Pump fluence dependence.** Normalized $\Delta R/R$ dynamics (top left) and spectra (top right) measured at A_{1s}/B_{1s} excitonic resonances at different pump fluences in the range 1-20 $\mu\text{J}/\text{cm}^2$ with pump photon energy of 2.75eV. The peak of A and B exciton bleaching signal displays a linear dependence with the pump fluence in the examined range (bottom left). The transient spectra at different time delays (bottom right) show the spectral profile of A and B excitons in the temporal region of explored in the experiment.”.

3. The authors show in Fig 4 that the fast decay clearly observed for low photon energies arises from the coherent contributions to the differential reflectivity.

a. If these are coherent contributions, wouldn't an experiment examining the degree of linear polarization, or degree of circular (valley) polarization, better support this claim?

Reply: We thank the Reviewer for the suggestion and we agree with her/him that a possible experiment aimed to examine the degree of valley polarization would better support this claim. However, valley polarized measurements introduce additional experimental difficulties with respect to the experiments here reported, because they require the use of sub-20-fs circularly polarized light pulses which are challenging to generate due to the requirement of broadband waveplates and polarizers. Right now, we are not able to produce sub-

20-fs circularly polarized pulses in order to study valley dependent exciton dynamics of these materials with the same temporal resolution achieved with linear polarization.

Action taken: We added the following sentence to the manuscript: “It is worth noting that, in this excitation regime, coherent exciton contribution could be more clearly disentangled from the incoherent exciton dynamics by performing pump-probe measurements with circularly polarized pulses and polarization resolved detection schemes. However, the use of broadband polarization optics makes difficult to preserve the extremely high temporal resolution need to observe such a process”.

b. In Fig S6 shows that this fast decay is only observed for cross-polarized pump and probe. This is opposite the expectation for a coherent process in 2D TMDs. Please explain.

Reply: We thank the Reviewer for noticing this issue. We accidentally swapped the labels of the polarization in Fig.S7. We apologize for this mistake. The probe polarization is vertical, and not horizontal. Indeed, for both A and B excitons, the dynamics reported in Fig. S7 for the parallel polarization configuration coincide with the ones reported in Fig. 2c in the main text, where both pump and probe beams are set linear and parallel as correctly reported in the Methods section.

The observed coherent contribution to the differential reflection results from the coherent effect of adiabatic following: the probed, pump-induced exciton density adiabatically follows the pump pulse shape which is centered at zero time delay [L. Allen, J.H. Eberly. Optical resonance and two-level atoms. (1987)]. This effect occurs for a sufficient off-resonant detuning between pump and probe pulses and leads to an ultrafast build-up of the differential reflection signal. From this off-resonant effect, one has to distinguish the formation of incoherent contributions which occur with a delay due to the additional exciton-phonon scattering processes. Our numerical evaluations show that coherent excitons dominate the ultrafast response during the pulse, whereas incoherent contributions to the differential reflection describe the incoherent scattering process, i.e., the signal after the polarization to population transfer is completed.

Action taken: We corrected the caption of Fig. S7. We have clarified the discussion and interpretation of our results in the manuscript and included the coherent signals as separate curves in Fig. 4. See the revised version of Fig. 4, here reported as Fig. R8.

Fig. R8: Coherent and incoherent exciton dynamics. a) Calculated exciton bleaching dynamics at the energy of the A_{1s} resonance considering only the coherent exciton contribution (shaded areas), only the incoherent exciton contribution (dashed lines) and both the contributions (continuous lines) for low (orange traces) and high (blue traces) pump photon energy, i.e. 2.29 eV and 2.75 eV, respectively. b) The same for the dynamics at the energy of the B_{1s} resonance.

4. Much of the validity of the analysis hinges upon interpreting the changes in reflected light intensity as due to Pauli Blocking. While the authors support this claim with comparisons to integrated spectra in Fig S2, there are marked differences in prior published works that require some comment and explanation.

a. The authors of this work have a prior publication on MoS₂, Pogna ACS Nano v10 p1182 '16, demonstrating that Pauli Blocking cannot explain the transient absorption spectra at a delay of 300fs after excitation. However the results here show no spectral shifts until after 1ps. Please explain why such different results are reported here.

Reply: We thank the Reviewer for the insightful comment. We believe that when the Reviewer mentions the spectral shifts, he/she is referring to what in the paper by Pogna et al. (Ref. [12]) are the photoinduced absorption (PIA) signals. These signals have opposite sign with respect to the exciton bleaching signals and they are red-shifted in energy with respect to the excitonic resonances. In the present paper, we detect both the A and B bleaching positive signals which have the same dynamics as the ones in Ref. [12] (note that the exciton bleaching signals appear to be negative in Figs.1-2 of Ref. [12] because instead of differential reflectivity maps we display differential absorption ΔA maps). In contrast, the PIA signals are strongly quenched in our data, as shown in Fig. S4 of the new version of the manuscript. We attribute the different intensities of the PIA signals to a photonic effect due to the different sample geometries. Pogna et al. measured in transmission on a MoS₂ sample deposited on a thick (100- μ m) SiO₂ substrate, while in our case the experiments are performed in a reflection geometry on MoS₂ deposited on a SiO₂(280-nm)/Si substrate. As the thickness of the SiO₂ layer is comparable to the visible wavelength, thin-film interference effects play a role and reshape the $\Delta R/R$ spectrum. Numerical simulations using the transfer matrix method support our results and will be the object of a separate publication. This is the main reason why the pump-probe spectra in the present paper are different with respect to ones previously measured in Ref. [12]. Other differences could be related to the fact that in the present case the sample is grown by CVD while in Ref. [12] it is mechanically exfoliated. As a consequence, the residual doping and the density of defects might be different and could result in slightly different broadening of the excitonic peaks and different relaxation dynamics on a long timescale. To be sure that the slowing down of the rise time at increasing pump photon energies is mainly a population-induced effect, we compared the $\Delta R/R$ traces measured at the maximum of the bleaching signals with the traces spectrally integrated around the excitonic peak (see Fig. S2). This operation allows us to remove the possible effect of the energy shift due to the photoinduced bandgap renormalization in the pump-probe spectra as explained in Ref. [2] of the SI. Since the same pump photon energy dependence is observed in the spectrally integrated traces, we can state that this effect is mainly driven by the exciton population dynamics, with other many-body effects playing a minor role.

Actions taken: We clarified some sentences in the main text and in the section 2 and 3 of the Supplementary Information and we explained why the $\Delta R/R$ spectra in this paper are slightly different from the ΔA ones reported in Ref. [12].

"The $\Delta R/R$ spectrum displays a symmetric profile around each excitonic resonance with no shift of the peak maximum, within the considered temporal window. Weak and broad negative features are detected at higher and lower probe photon energies as better shown in Supplementary Fig. S2. These features are strongly quenched with respect to previously published pump-probe measurements on 1L-MoS₂ performed in transmission geometry[refs]. We attribute the different intensities of these signals away from the excitonic transitions, to a photonic effect caused by the interference of multiple reflections of the incoming beam in the thin SiO₂ substrate".

We removed the sentence: "This means that, for a delay time < 1ps the non-equilibrium optical response can be simply described in term of a transient reduction of the exciton oscillator strength due to Pauli blocking effect." in the caption of Fig. S4.

We also modified the sentence in section 3 from "The observed transient behavior .." into "The fact that the $\Delta R/R$ is dominated by the bleaching signal while the transient signal away from the excitonic resonance is extremely weak strongly suggests that the population effect dominates over many-body processes."

b. Similarly, the differential reflectivity here, e.g. in Fig S3, differ significantly from past spectra: e.g. Sim et al. PRB v8 p075434 '13 and Borzda et al. Adv Funct Mater v25 p3351 (co-authors by some of the current authors), both of which report numerous positive and negative features which are either assigned to spectral shifts or charged excitons. Please explain why such different results are reported here.

Reply: Please see our extensive discussion of this point in the previous response. The negative features are strongly suppressed in the present configuration because of the interference effects occurring in the reflection geometry due to the thin SiO₂ layer.

REVIEWERS' COMMENTS:

Reviewer #1 (Remarks to the Author):

The authors have satisfactorily addressed all my concerns in the revised manuscript and supporting info. Now, I highly recommend its publication in Nature Communications. Congratulations!

Reviewer #2 (Remarks to the Author):

I find that the authors have satisfactorily addressed all of my comments. They have corrected a few mistakes/typos that otherwise raised possible contradictions between this work and others in literature. They have explained their reasoning on neglecting higher energy pumping, which is sound, and have included those result here anyway for the sake of clarity and full disclosure. They have also expanded their theoretical description to included dynamic screening effects, which has left their conclusions unchanged.

The results here are important and improve our understanding of exciton formation in TMDs. The work is now more clear and quite thorough. It is appropriate for the readership of Nat. Commun. and I support its publication without further revision

Reviewer #1 (Remarks to the Author):

The authors have satisfactorily addressed all my concerns in the revised manuscript and supporting info. Now, I highly recommend its publication in Nature Communications. Congratulations!

Reply: We thank the Reviewer for highly recommending publication of the paper in Nature Communications.

Reviewer #2 (Remarks to the Author):

I find that the authors have satisfactorily addressed all of my comments. They have corrected a few mistakes/typos that otherwise raised possible contradictions between this work and others in literature. They have explained their reasoning on neglecting higher energy pumping, which is sound, and have included those result here anyway for the sake of clarity and full disclosure. They have also expanded their theoretical description to included dynamic screening effects, which has left their conclusions unchanged.

The results here are important and improve our understanding of exciton formation in TMDs. The work is now more clear and quite thorough. It is appropriate for the readership of Nat. Commun. and I support its publication without further revision.

Reply: We thank the Reviewer for judging our results important and for recommending publication of the paper in Nature Communications without further revision.